# An AI-Powered System for Residential Demand Response

Iker Esnaola-Gonzalez [1,*], Marko Jelić [2,3], Dea Pujić [2,3], Francisco Javier Diez [1] and Nikola Tomašević [3]

1   TEKNIKER, Basque Research and Technology Alliance (BRTA), Iñaki Goenaga 5, 20600 Eibar, Spain; francisco.diez@tekniker.es
2   School of Electrical Engineering, University of Belgrade, Bulevar kralja Aleksandra 73, 11120 Belgrade, Serbia; marko.jelic@pupin.rs (M.J.); dea.pujic@pupin.rs (D.P.)
3   Institute Mihajlo Pupin, University of Belgrade, Volgina 15, 11060 Belgrade, Serbia; nikola.tomasevic@pupin.rs
*   Correspondence: iker.esnaola@tekniker.es

**Abstract:** Recent studies show that energy consumption of buildings has dramatically increased over the last decade, accounting for more than 35% of global energy use. However, with proper operation, significant energy savings can be achieved. Demand response is envisioned as a key enabler of this operation enhancement, as it may contribute to the reduction of demand peaks and maximization of renewable energy exploitation while mitigating potential problems with grid stability. In this article, a system based on artificial intelligence that solves the complex multi-objective problem to bring demand response programs to the residential sector is proposed. Through the application of novel machine learning-based algorithms, a unique control loop is developed to help dwellers determine how and when to use their appliances. The feasibility and validity of the proposed system has been demonstrated in a real-world neighbourhood where a notable reduction and shift of electricity demand peaks has been achieved. Concretely, in accordance with extreme changes in the energy prices, the users have demonstrated the ability to shift their demand to periods with lower prices as well as reducing power consumption during periods with higher prices, thus fully translating the demand peak in time.

**Keywords:** demand response; demand flexibility; artificial intelligence; machine learning; energy savings; peak shaving

## 1. Introduction

Buildings' energy consumption has dramatically increased over the last decade due to different factors including the population growth, the increase in time spent indoors or the increased demand for building functions and indoor life quality [1]. As a matter of fact, according to the United Nations Environment Programme (UNEP), buildings account for more than 35% of global energy use and nearly 40% of energy-related $CO_2$ emissions [2]. Nevertheless, significant energy savings can be achieved in buildings if they are properly operated.

In this domain, the residential sector is specially promising as it is characterized by many end consumers with relatively low individual energy demand, but with very high demand when considered in terms of home clusters, districts and residential communities. Evidence of this is that residential buildings represented the 25.4% of final energy consumption and 17.4% of gross inland energy consumption in the EU in 2016 [3]. The major end-uses responsible for these figures are space and water heating, followed by appliances, cooking and lighting [4].

Apart from the large energy consumption of buildings, peak energy demand certainly attracts lots of attention because of its negative impacts on energy grid capital, operational costs and environmental pollution to name a few. This impact is a direct consequence of the carbon-intense generation plants that grid operators deploy in order to satisfy energy demand during these peak periods [5]. One of the resources that could contribute to

significantly reduce peak demands are renewable energy sources (RES) [6], which are increasingly penetrating the energy production side. However, due to their intermittent nature, the renewable energy availability commonly does not match the distribution of energy demand in time, which may hinder their management and exploitation.

Demand-side management (DSM) activities including load curtailment (i.e., a reduction of electricity usage) and load reallocation (i.e., a shift of energy usage to other off-peak periods) have a huge potential as aids in matching energy demand with energy supply, thus avoiding these undesirable peaks. Furthermore, demand response (DR) programs are introduced into the smart grids so that reliable and economical operation of power systems are ensured [7]. DR can be understood as the set of technologies or programs that concentrate on shifting energy use short-term to help balance energy supply and demand [8]. In combination with energy generated from RES, DR is envisioned as one of the crucial enablers of curbing energy demand peaks [9].

However, the implementation of DR programs is not straightforward. The main barriers to adopt DR programs include regulatory, economic, technological and social issues [10]. Furthermore, solving the energy demand optimization for residential neighbourhoods that takes into account management of collectively shared energy assets such as RES generation as well as variable pricing tariffs and specific demand flexibility constraints is a complex multi-objective problem that requires the utilization of artificial intelligence (AI) systems. And this is where the RESPOND H2020 project (https://project-respond.eu, accessed on 15 March 2021) originates, aiming to bring DR programs to neighborhoods across Europe. With most contemporary DR implementations considering only the industrial sector, this paper proposes the simultaneous utilization of multiple different AI-based technologies such as machine learning and optimization to assist residential consumers that would like to make use of DR and incorporate it into their energy management systems. The proposed system considers the forecasted energy production and consumption based on the data acquired by the deployed Internet of Things (IoT) equipment and looks for modifications that would mitigate potential instabilities in the energy supply network by applying optimal energy utilization and load shifting. Finally, the predicted and optimal behavior are analyzed and corrective actions are suggested for dwellers, with the system even allowing for the possibility of semi-automated (remotely actuated with individual dweller consent) and fully automated (remotely actuated with previous persisting dweller consent) actions to be taken in order to minimize the additional effort that needs to be undertaken and that would disrupt daily habits. In summary, the main contributions of the article are the following:

- The development of an AI-powered system for demand response in residential buildings;
- The application of the AI-powered system in a real-world neighbourhood;
- Successfully validating dwellers' willingness to adjust their loads with regards to changing energy price tariffs in a real-world neighbourhood while noting:
  - Decrease of energy demand peaks by above 30%;
  - Increase of energy demand in non-peak periods by nearly 50%;

The remainder of the article is structured as follows. Section 2 analyzes the relevant previous work related to the presented problem. Section 3 introduces the RESPOND project. Section 4 presents RESPOND's AI system for optimal energy dispatching with DR events in ind for avoiding energy demand peaks and maximizing energy coming from RES while Section 5 showcases it with a real-world example. Finally, the conclusions of this work are presented in Section 6.

## 2. Related Work

Recent research work in the field of domestic energy use has been focused on removing the assumption that the demand is given and fixed, and on investigating feasible DSM approaches that dynamically adjust the demand, in order to fulfil or improve a specified performance requirement. These methods yield additional flexibility and introduce new degrees of freedom as part of novel energy management approaches and commercially

available products (e.g., Siemens DRMS or Akuacom DRMS by Honeywell), since the demand had otherwise been assumed to be passive and static in classic formulations. In this regard, a number of approaches investigating the demand-side optimization have been analyzed. For example, a multi-objective genetic algorithm approach for implementing DSM activities in an automated warehouse has been presented [11]. Furthermore, a modified genetic algorithm has been used to optimize the scheduling of direct demand control strategies [12]. An autonomous and distributed demand-side energy management system based on game theory has also been proposed [13]. Additionally, an autonomous DR system that tries to achieve both optimum and fairness with respect to the involved participants was designed in [14]. A fuzzy logic approach utilizing wireless sensor networks (WSN) and smart grid incentives for load reduction in residential heating, ventilation and air conditioning (HVAC) systems has also been presented in [15]. Finally, an integration of RES and electric vehicles with proper home DSM has been evaluated through different scenarios in [16].

Integrated approaches that considers both the supply side through optimal energy dispatching and demand-side through and DSM have, in general, received somewhat less attention in the literature. An integrated DSM program for multiple entities (represented by designated Energy Hubs) was proposed as a non-cooperative game within a cloud-based infrastructure in [17]. This DSM program was demonstrated only for entities with critical loads hence optimizing only their supply side. Each entity was incentivized to participate in the program which required the exploitation of different supply energy carriers, thus affecting the overall energy supply price value. This approach did not consider the possibility of influencing the non-critical demand, which may be key to exploit the full capabilities of DSM approaches. The Energy Hub concept has also been applied for the optimization of energy flows in simulated interconnected networks [18], but without taking into account DSM actions.

Another existing problem in the scenario presented in this article is accurately forecasting the energy to be produced by RES and to be consumed by the dwellers. Energy production forecasting approaches can be divided in three groups depending on the approach used for the estimation of the production: physical models, statistical models and hybrid models [19–21]. However, what all of these methodologies have in common are their inputs, as they all model the dependency of the renewable production on the weather conditions. Physical approaches were presented first and offer models represented by sets of mathematical equations and physical laws which depict the renewable system. Even though they were replaced by the novel data-driven approaches in the field of, for example, photovoltaic (PV) panel energy forecasting, these models are practically the only ones presented in literature regarding solar thermal collectors (STCs) production [22]. However, for PV forecasting, physical models are usually outperformed by data-driven approaches, which are present in state of the art (SoA) articles. Nonetheless, due to the fact that their estimation is based on the mathematical modeling of the system, their main advantage is that they do not need any historical data, so in some scenarios in which historical data is not available, they are the only applicable ones. However, for the application of these methodologies, numerous physical parameters are required. This is a significant drawback, having in mind that these characteristics are usually hard to access. On the other hand, data-driven models, both regressive (autoregressive (AR), autoregressive moving average (ARMA), autoregressive integrated moving average (ARIMA), autoregressive moving average model with exogenous inputs (ARMAX), nonlinear autoregressive moving average with exogenous inputs (NARMAX), etc.) and machine learning-based (neural networks, support vector machines, random forests, kNNs, etc.) require a large amount of historical data, but are capable of much more precise modelling, which significantly improves performances. Additionally, none of the physical parameters are required in order to implement this approach. Finally, hybrid approaches tend to combine benefits from previously presented models in order to further improve upon them.

Similar to the energy production forecasting, there is extensive research on the topic of forecasting of energy demand. One study investigates fifteen anonymous individual household's electricity consumption forecasting using a support vector regression (SVR) modelling approach, applied both to daily and hourly data granularity in [23]. In this experiment, households' occupation, dwelling properties and socioeconomic status were unknown. Therefore, aggregating hourly consumption to daily was an effective way to mitigate the impact of randomness in hourly behaviours of family members. Under the assumption that there usually exists an intrinsic low-dimensional structure governing the data recorded from a collection of residential houses and that using this structure in load forecasting can help improve the forecasting performance, a compressive load forecasting approach incorporating both temporal and spatial information is presented in another study [24]. The proposed method is called nonuniform compressive spatio-temporal load forecasting (CST-LF) as it is inspired by compressive sensing (CS) and structured-sparse recovery algorithms, and it is tested against various benchmark models using real and high-quality data, showing that the proposed approach improves the short-term electric demand forecasting. A research focused showing how calendar effects, forecasting granularity and the length of the training set affect the accuracy of a day-ahead load forecast for residential customers [25]. Regression trees, neural networks, and support vector regression were tested, and the former was the technique obtaining best results. The use of historical load profiles with daily and weekly seasonality, combined with weather data, leaves the explicit calendar effects a very low predictive power. In the setting studied in the article, it was shown that forecast errors can be reduced by using a coarser forecast granularity. It was also found that one year of historical data is enough to develop a load forecast model for residential customers as a further increase in training data set has a marginal benefit.

When considering optimization approaches, various applicable methodologies can be found in the related literature. Some authors employ complex nature-inspired heuristics such as the genetic algorithm as in [26,27], particle swarm optimizations as in [28], as well as artificial neural networks like in article [29]. However, when working with data with medium-sized resolutions such is the case with hourly measurements that are most often given by RES production and demand forecasting algorithms, more efficient algorithms with simplified models such as linear programming and its extension, mixed-integer linear programming (MILP), are used more often. Day-long optimizations using a MILP model with PV and storage systems are detailed in [30], with the same horizon also found in [31,32] with a larger temporal resolution (15 min). On the other hand, MILP models have also been employed for long-term feasibility assessments owing to their high efficiencies, as analyzed in [33]. However, this application of optimizations is out of the scope of this paper.

## 3. The RESPOND Project

The RESPOND project aims to deploy and demonstrate an interoperable, cost-effective and user-centered solution, entailing energy automation, control and monitoring tools for a seamless integration of cooperative DR programs into the legacy energy management systems. In this endeavor, RESPOND leverages an integrated approach for an optimal energy dispatching, taking into account both supply and demand side, while exploiting all energy assets available at the site. More specifically, RESPOND aims at reducing energy demand peaks and maximizing the exploitation of renewable energies, by implementing a set of control actions that cause dwellers as few disturbances as possible in their everyday life.

Towards that goal, a central IoT platform has been developed for the acquisition, processing and exploitation of relevant data collected in neighborhoods. This platform is comprised of different components and its architecture is depicted in Figure 1, with more details presented in Section 4.1. It is worth mentioning that ensuring consumer's data privacy has been a prime requisite of this platform, since consumers are often concerned with sharing their energy consumption data as found in [34]. As a matter of fact, a formal specification of data usage requirements for the built environment has been proposed towards the development of a trustworthy data sharing ecosystem in [35].

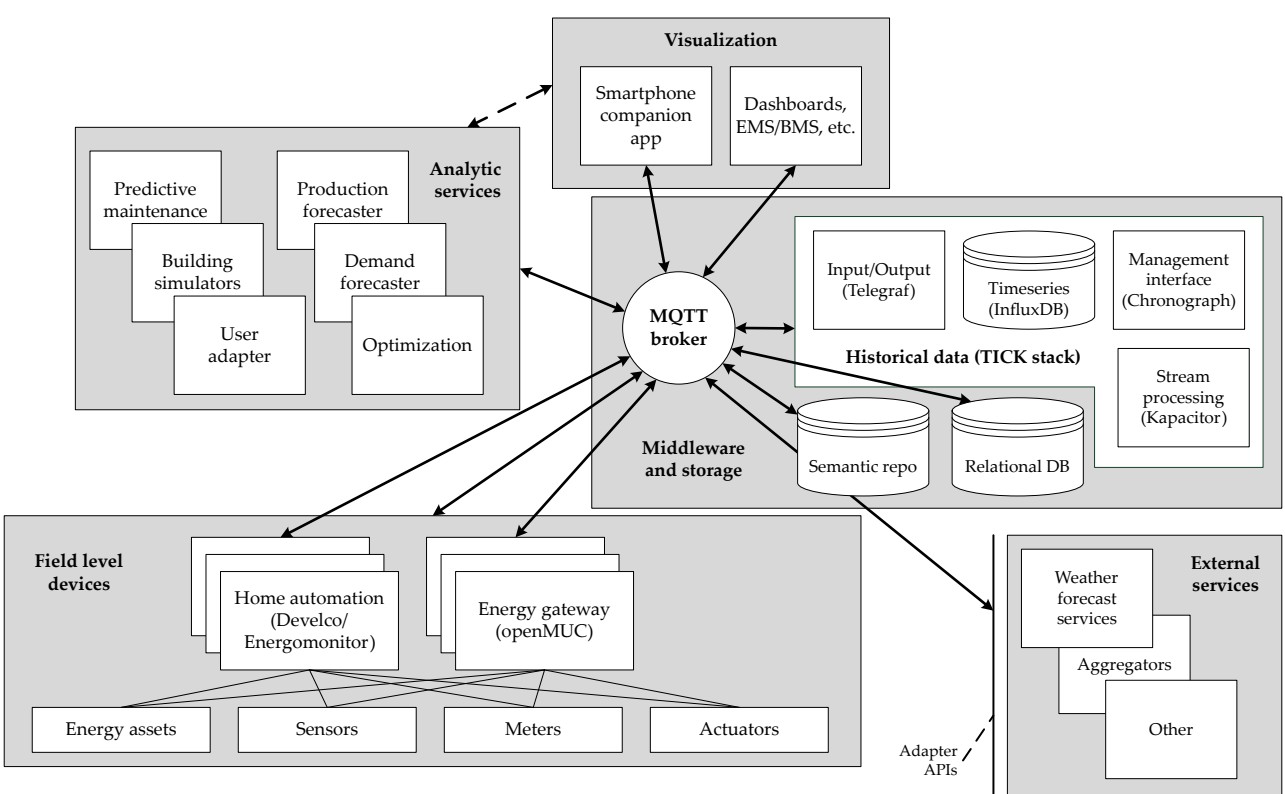

**Figure 1.** The RESPOND Internet of Things (IoT) platform architecture for acquiring, processing and exploiting data.

With the purpose of demonstrating the RESPOND solution, it is being implemented in different types of residential buildings (i.e., apartments, single-family and multi-family houses), situated in different climate zones (i.e., Mediterranean, oceanic and humid continental climate), but sharing the same microclimate within each pilot site, having different forms of ownership (i.e., rental and home-ownership), population densities and underlying energy systems. Namely, the three RESPOND pilot sites are located in Aarhus (Denmark), the Aran Islands (Ireland) and Madrid (Spain). Each pilot site is regarded as a neighbourhood of dwellers that can make use of some type of grid connection (for either electric energy, heating or both) alongside a locally generated renewable source. In that sense, each pilot provides a suitable testbed for the integration of DR management technologies. Since DR has historically been significantly more prominent in the industrial sector, and with residential users presenting unique challenges, a set of AI-based services is foreseen by the RESPOND project to help users make the best use of different types of available energy while also providing positive effects for the distribution system through the utilization of demand-side flexibility. These services include machine learning-powered forecasting for both energy generation and energy consumption coupled with an optimization algorithm that is supposed to provide feedback to the dwellers regarding the best way in which to organize their consumption in different conditions.

Concretely, the residents of the Aarhus pilot can make use of either grid imported electricity along with locally produced energy via PV panels to fulfill their electric demand. On the other hand, they rely on a district heating system for their thermal and domestic hot water demand. Likewise, besides grid connections, the dwellers on the Aran Island pilot also have locally produced electricity via PV panels while they fulfill their thermal domain using heat pumps. The Madrid site has a shared solar thermal collector which, along with two gas boilers, fulfils the heating demand while electric demand is met using only the grid connection.

Having such a heterogeneous group of dwellers hinders the diffusion and impact of DR solutions and makes it more difficult to ensure sustained user engagement with

DR programs. This is why interaction with dwellers is considered as a key point in the RESPOND project. Consequently, as previously mentioned, a set of tools and services are planned to deliver measurement-driven suggestions to dwellers for energy demand reduction and influence their behavior making them an active indispensable part of DR loop. One of these tools is the RESPOND mobile app, a multilingual and cross-platform mobile app which contributes to the user engagement. It is available for download both in Google Play and App Store and gives the dweller direct and detailed insight into all relevant monitored data. The added value of the RESPOND mobile app lies in its ability to suggest to dwellers energy conservation opportunities, which are a direct result of the AI system described in the next Section.

## 4. An Artificial Intelligence System for Optimal DR Strategies

Although AI is not something new, currently it is experiencing an upsurge that can be attributed to advances in computing and the increasing availability of data [36]. Different definitions for AI can be found in the literature and according to the European Commission's High-Level Expert Group on Artificial Intelligence [37], AI systems are software (and possibly hardware) systems that, given a complex goal, act in the physical or digital dimension by perceiving the environment through data acquisition, interpreting the collected structured or unstructured data, reasoning on the knowledge or processing the information derived from this data and deciding the best action(s) to take to achieve the given goal.

RESPOND aims to allocate the most suitable demand profiles both at a dwelling and neighborhood levels as a driver for reducing the energy demand in specific time periods, as well as for maximizing the exploitation of renewable energy. To do so, RESPOND has proposed the AI system depicted in Figure 2 which has five main blocks: measurement, forecasting, demand response message generator, optimization and control block. In briefest possible terms, the measurement block is in charge of collecting and storing all the available sensor and non-sensor data. The forecasting block attempts to combine records of this data to provide projections of future production and consumption. Having in mind global issues such as demand and production peaks, a grid-responsible entity is given an opportunity to define so-called DR messages that are used as inputs to the optimization block (along the forecasted profiles) in order to guide the model towards the desired load curve. Finally, the outputs of the optimization block are analyzed by its control counterpart which suggests concrete control actions that the dwellers (end users) should carry out. With more detailed explanations of each component given in the following paragraphs, through the adequate interaction between these services, optimal dispatching of energy with regards to DR events is ensured.

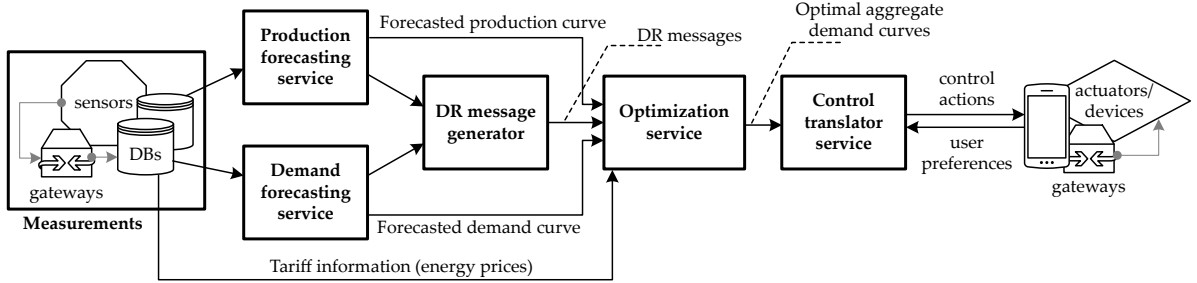

**Figure 2.** The RESPOND Artificial Intelligence (AI) system loop.

Next, each of these five blocks are detailed. Although the proposed AI system is developed with flexibility in mind to handle different types of energy carriers including electric, thermal and domestic hot water (DHW), as well as their possible interconnection points like converters and heat exchangers, the demonstrations given in following sections

as well as the use case elaborated on towards the very end of the paper will focus solely on the electric domain.

### 4.1. Measurement

The sensor technology embedded in IoT devices is continuously becoming cheaper, more advanced and more widely available, thus moving beyond disruption to become a mainstay of daily life. The residential sector is no exception to this expansion, and according to a report conducted by Navigant Research [38], the global annual revenue from residential IoT device sales will reach $167.2 billion in 2027.

In the context of the RESPOND project, the monitoring of real-world qualities and events within dwellings and neighborhoods is performed with smart home monitoring solutions provided by Energomonitor (https://www.energomonitor.com, accessed on 15 March 2021) and Develco (https://www.develcoproducts.com, accessed on 15 March 2021). These solutions comprise the equipment necessary for the acquisition of observed ambient (e.g., temperature or humidity) and energy (e.g., electric demand or gas consumption) data. Regarding the energy consumption, it was registered both at an appliance level with a smart plug (which also allowed remotely switching it on or off) and at a household level with smart meters. Monitoring is complemented with an OpenMUC (https://www.openmuc.org, accessed on 15 March 2021)-based gateway which is also able to acquire data from other monitoring and control applications that are not from Energomonitor or Develco. Regarding other relevant data such as energy price or weather information, which is not monitored with installed physical devices, it is collected from external sources. As illustrated in Figure 1, all this data is sent to the MQTT (Message Queuing Telemetry Transport) broker middleware, which allows the integration of information coming from different sources, as well as the communication between different components. The communication with the MQTT broker is done via the publish/subscription method, which decouples the client that sends the message (the publisher) from the client or clients that receive the messages (the subscribers). Since there are differences in hardware and software implementations of devices produced by different vendors, a canonical data model (CDM) is designed to work along with the MQTT message exchange protocol and to ensure interoperability among different system components. The implementation of the CDM enables unifying the data using protocol converters that abstract diverse protocols of legacy and newly installed equipment. The benefit of such an approach is that each protocol needs to be converted only into common format and back, which results in a linearly growing number of adapters ($2N$) compared with the traditional exponential growing number of adapters ($N^2$).

Complimentary to the data acquisition equipment and services, the RESPOND Measurement block counts on data repositories. Due to the diverse types of data collected, three different database systems are considered—time series databases (TSDB), triplestores or semantic repositories and relational databases.

IoT data, which is characterized by its abundance, is recommended to be stored in TSDBs. These databases are optimized for time series data and designed to handle high write and query loads as well as down-sampling and deletion of old data, thus being able to manage an amount of data while ensuring a high performance. This is why, one of RESPOND's data storage systems is InfluxDB (https://www.influxdata.com/products/influxdb-overview, accessed on 15 March 2021), an open source TSDB.

In the built environment, the integration of static building information and IoT data has become one of the main challenges [39]. Furthermore, easy and intuitive ways to rapidly browse, query and use building information combined with IoT data are not usually available [40]. Semantic technologies can aid in solving these issues, as they allow for more dynamic manipulation of the building information in resource description framework (RDF) graphs by means of query and rule languages. Therefore, the RESPOND platform takes advantage of a semantic repository to store the static building information. Semantic repositories are optimized for hosting this type of data and usually support a

SPARQL endpoint where data can be queried using SPARQL queries. Namely, an Openlink Virtuoso (https://virtuoso.openlinksw.com, accessed on 15 March 2021) repository is used. Both practice and research suggest the use of a graph-based format to capture building data, nevertheless keeping numeric data explicitly out of the semantic graph for computational performance reasons [41] and this is the approach followed by RESPOND [42].

Last but not the least, structured data that is massively instantiated but is not time-based is stored in a relational database. Namely, RESPOND's relational database system is MySQL and the results of the forecasting block, the DR message generation block and optimization block, depicted in the following sections, are all stored in the MySQL database so that they can be queried asynchronously by other services and RESPOND mobile app.

### 4.2. Forecasting

Once the collected data is stored in the adequate data repositories, it remains accessible to be exploited for different purposes. Some of this data, such as the monitored electric consumption of appliances and the dwelling as a whole, is used for visualization purposes in the RESPOND mobile app. The stored data can also be exploited by analytic services, which are the core of this Forecasting block.

Being able to accurately predict the amount of energy to be produced over a period of time and knowing in advance when demand peaks will occur, can definitely contribute to better management of their disparity, thus allowing the suggestion of the most suitable actions to dwellers. Therefore, in this block, two main services are considered: the RESPOND Energy Production Forecasting Service and the RESPOND Energy Demand Forecasting Service.

#### 4.2.1. RESPOND Energy Production Forecasting Service

As a part of the RESPOND project, two different RES are identified—PV panels installed in Aarhus and the Aran Islands, and STCs in Madrid. Therefore, three different day-ahead energy production forecasters with hourly time resolution were developed.

The crucial motivation for developing various models for RES production forecasting corresponds to their stochastic nature, most evident in high correlation between the energy they produce and meteorological conditions. Therefore, day-ahead hourly weather forecasts for various factors were obtained from Weatherbit (https://www.weatherbit.io/, accessed on 15 March 2021), with the solar irradiance being the most important one. Namely, the correlation between solar irradiance and energy production is expected to be extremely high, which is why it was crucial to include it as an input for the forecaster. Apart from the irradiance, UV, wind direction and speed, outside temperature, cloud coverage and relative humidity were also considered as relevant inputs.

Current state-of-the-art solutions for PV production forecast modeling are mainly focused on various machine learning approaches [43,44] as they achieve the highest performances when there is ample available data. Unlike the Aarhus pilot site where two larger buildings are sharing a single PV plant and historical data for a period of 2 years was available, making it possible to proceed with the machine learning approaches, the Aran Islands pilot consists out of different, geographically separated, houses with each of them having its own PV production. Furthermore, for more than half of the participant dwellings, no production measurements were available, which is why the traditional physical model approach has been chosen for the Aran Islands pilot [45]. Conversely, for STCs, even though physical approaches are the most frequent ones [22], due to the fact that production measurements were recorded and stored using the RESPOND platform, a machine learning approach has been applied here. To sum up, for two, our of the three pilot sites, Aarhus and Madrid, machine learning models have been used, whilst in the third one, in Aran Island, Ireland, due to the lack of data, physical models were exploited.

Various machine learning approaches were considered and tested for Aarhus and Madrid pilot sites, such as support vector regression, linear regression, different neural network architectures, kNN and random forecast. For each of them, several different

hyperparameters have been testes. For support vector regression, defined to solve dual problem defined as

$$\min_{\alpha,\alpha*} \frac{1}{2}(\alpha - \alpha*)Q(\alpha - \alpha^*) + \epsilon e^T(\alpha + \alpha^*) - y^T(\alpha - \alpha^*)\Bigg|_{e^T(\alpha-\alpha^*)=0 \text{ and } 0\leq\alpha_i,\alpha_i^*\leq 1/\lambda, \ i=1,...,n},$$

with $\lambda$ being regularization parameter, $e$ vector of all ones, $Q$ positive semidefinite matrix $n$ by $n$, where $Q_{ij} = K(x_i, x_j)$ and $K(x_i, x_j)$ kernel, various regularization factors and kernels (e.g., linear, sigmoid, radial basis) together with the corresponding parameters were tested. In case of linear regression, where goal is minimizing following criterion

$$J = (xW - y)^T(xW - y) + \lambda|W|,$$

with $\lambda$ being regularization coefficient, polynomial degree of the input $x$ and regularization coefficient have been optimized. Furthermore, when neural networks were considered, number of hidden layers and corresponding number of neurons were optimized. In case of kNN algorithm, which calculates the distance $d_i$ between the training sample $X_i$ and current input $X$, and reorder them so that

$$d_{l_1} \leq d_{l_2} \leq \cdots \leq d_{l_n},$$

in order to provide the output as the mean of the training outputs of the $k$ samples which correspond to the smallest $k$ distances, number of optimal neighbours $k$ has been tested. Finally, for random forecast algorithm, which estimates the output as the average of the individual predictions of each tree

$$y = \frac{1}{l}\sum_{i=1}^{l} f_i(x),$$

where $l$ is number of trees and $f_i$ estimation of the $i$-th tree, polynomial degree of the input, number of trees $l$ and their maximal depth has been optimized.

In all cases, optimal hyperparameters were chosen using grid search and the mean absolute percentage error (MAPE) was used as an indicator of their performance. For forecasting energy coming from PV panels, the best models, out of 935 tested, with the lowest MAPE were random forest models with 50 estimators, whilst for the STC the neural networks with 2 hidden layers containing 40 and 5 neurons, respectively performed the best, with more details given in [46]. All of the aforementioned models, together with a whole production service as a whole, have been developed, trained, tested and compared in Python, using Keras (https://keras.io/, accessed on 15 March 2021) and sckit-learn (https://scikit-learn.org/, accessed on 15 March 2021) libraries.

For the Aran Islands, the physical model presented in [47] was employed, as it required only parameters that can most commonly be found in the corresponding PV cells data sheets. The main concept of this methodology is to estimate the final production using the cell temperature, which is estimated using two groups of input parameters: proprietary PV cells static parameters (longitude, latitude, time zone offset, slope of the PV cell surface, rated capacity of the PV array, temperature coefficient, surface area of the PV cell, nominal operating cell temperature) and dynamic ones (global horizontal radiation, ambient temperature, number of the day in the year, cloud coverage, current time). Hence, for each Aran house PV array, static parameters were obtained and models were developed to estimate production of each household.

Finally, after training machine learning models and the development of the physical one, they have been tested and the obtained MAPEs are as follows: 8.3% for Aarhus model, 21% for Aran and 6.2% for Madrid. For each of the pilot sites, the MAPE has been calculated on the neighbourhood level which means that the aggregate profile for all apartments/houses was provided. For the MAPE calculation, a special batch of the

testing data originating from the pilot sites, has been used. Depending on the pilot and the corresponding data availability, the duration of the window that has been considered varies, but it covered from two to six months with different seasons in all three cases included. Figure 3 illustrates the comparison between the real and the forecasted PV power generation in Aarhus as an example of estimator performances.

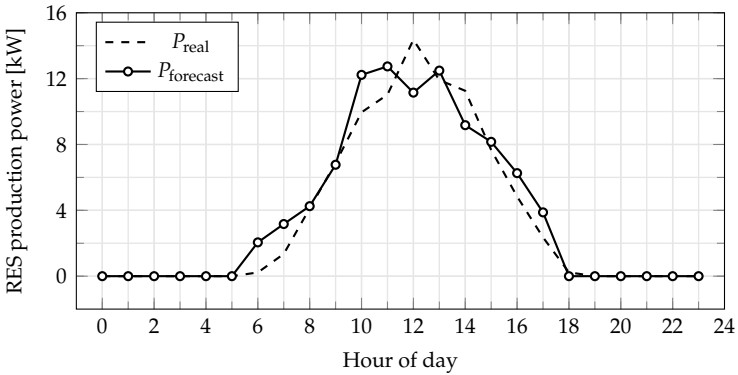

**Figure 3.** RESPOND production forecasting service output performance for an Aarhus photovoltaic (PV) array.

### 4.2.2. RESPOND Energy Demand Forecasting Service

Alongside with the forecasting of the energy produced from RES, the forecasting of the energy demand is essential in RESPOND to allow the dispatching of optimal DR strategies. This service aims at forecasting short-term energy demand, that is, the energy to be consumed during the next 24 h with an hourly frequency. Focus is placed on the electric demand at a house level, therefore, a model is built for predicting the short-term electric consumption of each dwelling participating in the RESPOND project. Afterwards, these predictions are aggregated to estimate the neighborhood demand prediction, as it is considered to be an an effective way to mitigate the impact of randomness in the behavior of different dwellers as found in [48].

This service is based on data-driven predictive models and machine learning algorithms and exploits the data previously collected in the Measurement block. It implements a multi-step ahead time series forecasting method, which consists in estimating the next $h$ values $\{y_{T+1}, ..., y_{T+h}\}$ using the previous values $\{y_1, ..., y_T\}$. Namely, it uses a *Multi-Input Multi-Output (MIMO)* Strategy [49].

As for the collection of machine learning algorithms that were tested for the development of this service, it included auto regressive integrated moving average (ARIMA), linear regression, support vector regression and K-Nearest Neighbor (kNN). A set of explanatory input variables were extracted from the time variable. The rationale behind this decision lies, on the one hand, in the simplicity and explainability provided, and on the other, in the possibility to impute missing values in the case of sensor failures. These variables were the day of the month, the month, the hour, the season, the day of the week and a Boolean variable indicating whether it is a working day or not. Some of these variables such as the month and the hour, have a cyclical meaning that is not reflected in the calculation of distances, therefore, a trigonometric transformation was performed (https://www.avanwyk.com/encoding-cyclical-features-for-deep-learning/, accessed on 15 March 2021). As a consequence, two variables were obtained from each transformation: one from the sine and the other from the cosine. Furthermore, before developing the predictive models, some outlier values were detected and removed.

The RMSE (Root-Mean-Square Error) values obtained with the ARIMA and SARIMA models were the highest ones amongst the tested ones. The linear regression lowered this error, although the coefficient of determination ($R^2$) was less than 0.3 in all the fitted models. The support vector machine-based models lowered even more the forecasting errors, but their computational cost and the fact that it forecasted electric consumption

values below 0 Wh made it inadequate for the service sought. The best fit was obtained with the kNN algorithm, and furthermore, the optimal *k* hyperparameter value was less than 5, thus improving its performance. Therefore, the RESPOND Energy Demand Forecasting service has been implemented based on the kNN algorithm. A more detailed explanation of the whole experimentation is provided in [50]. Figure 4 illustrates the comparison between the real and the forecasted energy consumption in a Madrid dwelling.

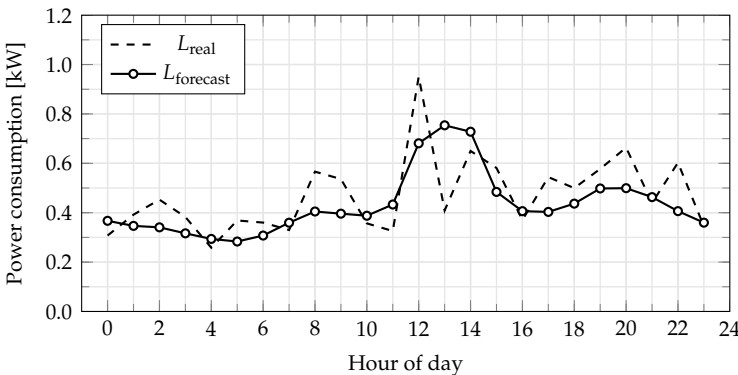

**Figure 4.** RESPOND energy demand forecasting service output for a dwelling in Madrid.

Under normal circumstances, the performance of forecasting models degrade over time due to a change in the environment that violates the models assumptions [51]. However, the COVID-19 pandemic situation has accentuated this degradation. As a matter of fact, the daily electric consumption habits of the RESPOND participants have changed because of the obligation to remain confined due to the state of alarm and restriction measures. In order to mitigate this performance and ensure that the demand forecast was accurate enough, a retraining and adaptation strategy has been implemented [52].

All the developed predictive models within the Forecasting block were developed in R programming language, using the functions within the caret (https://cran.r-project.org/web/packages/caret, accessed on 15 March 2021) package. They were later exported in the form of *.rds* files and deployed in an R server within the RESPOND platform and their results were stored in a MySQL database. Here, they remained available for their visualization in the RESPOND mobile app, as well as for their further exploitation by the Optimization block.

### 4.3. Demand Response Messages

As previously mentioned, DR implementations have already existed in the industrial sector for some time. The concept, with minor modifications, essentially adheres to the following structure: an aggregator or similar intermediary entity makes arrangements with industrial users that can offer flexibility in terms of utilizing a set of large consuming processes or machines. The frequency, duration and intensity of allowed load modifications coming from these consumers is contracted, meaning that the industrial user is expected to, upon request or at certain predefined times, allow for power required by the mentioned processes and machines to either be significantly decreased or increased. This mechanism is most commonly used for load reduction with large-scale heaters and chillers for easing the burden on the grid during peak hours. However, similar mechanisms can also be employed to balance unexpected production spikes, especially in systems that incorporate large capacities of renewable generation. In return for the provided flexibility, the aggregator offers monetary compensation while it also receives compensation from the grid-side operators for improving the stability of the system.

On the other hand, despite its notable potential contribution [53], the penetration of DR programs into the residential sector has been relatively slow, with very few business expressing interest in this aspect. As a result, standards that define how the necessary load modifications should be formulated for domestic users are arguably scarce. Hence,

the methodology adopted by the RESPOND AI system offers flexibility in terms defining these, so-called, DR messages. These messages are envisioned to be transmitted by a grid-responsible entity (e.g., aggregator) and sent to the system so that it knows in which way it would be beneficial for the grid to reshape the load. Using these so-called messages as guidance, the optimizer will derive the optimal profile, and afterwards, concrete control actions will be suggested for dwellers and their appliances. Since there are different ways in which an aggregator could direct the system towards the necessary changes, two types of DR messages are considered: implicit and explicit.

### 4.3.1. Implicit DR Messages

The term implicit DR is used within the context of the RESPOND AI system to refer to instances where the optimal demand profile is inferred without specifically defining the amounts by which the load should be modified at different times. As opposed to that approach, indirect constraints are provided that guide the model towards the desired behaviour. One example of such DR applications are cases where the dwellers are offered tariffs from their respective energy service companies (ESCOs) that utilize variable energy prices. The most basic and widespread example of such cases are static time of use (ToU) tariffs that offer lower prices at certain periods of the day. This is done in order to facilitate the translation of some portion of demand to hours with cheaper energy, so that the strain on the grid during peak times is reduced. In general, ToU tariffs offer lower prices during the nighttime and sometimes mid-day periods where most of the dwellers are either sleeping or at work, so the aggregate demand is generally lower compared with other periods of the day. Hence, these periods are the most suitable to be occupied by large consuming appliances (e.g., washing machines, dryers, dishwashers, heaters, boilers, etc.) that have activations which can be shifted in time. However, there are also more extreme cases that include critical peak rebate (CPR) and critical peak pricing (CPP) periods where the energy costs are respectively extremely low or extremely high, thus allowing for more specific moulding to be performed on the load profile. Such a case was implemented for a limited time to test the effects of a mixed tariff in the Madrid pilot site with the goal of observing dweller responses to so called "happy hours" with free energy. In order to illustrate typical examples for ToU tariff profiles, Figure 5 presents two cases where (a) shows static ToU prices that are offered with the Megawatt 2.0 DH tariff by Fenie Energia during the summer period and (b) shows the a more extreme CPR/CPP mixed tariff.

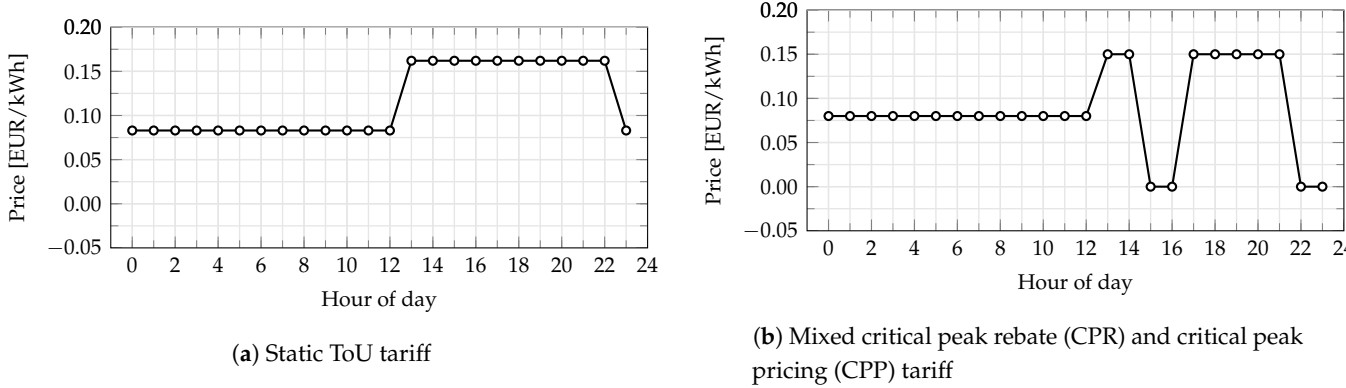

(**a**) Static ToU tariff

(**b**) Mixed critical peak rebate (CPR) and critical peak pricing (CPP) tariff

**Figure 5.** An example of time of use (ToU) tariffs that can be utilized for implicit demand response (DR).

### 4.3.2. Explicit DR Messages

Another option for steering the demand towards the desired values is to explicitly define the necessary corrections. Namely, since the system includes both production and demand forecasting services, both the supply and demand curves are known before the optimization is activated. By analyzing their relation for any major disparities, or by considering them separately, an aggregator-like entity or balance-responsible party can,

using its expert inputs, determine precisely the demand that should be adjusted. In order to provide support for such mechanisms, the RESPOND AI system envisions so-called explicit DR messages that influence the optimization output. Namely, in accordance with the temporal resolution and horizon dictated by other services in the control loop, explicit DR messages adhere to a similar format as other variables in the system.

Table 1 illustrates an example of an explicitly defined request for demand alterations, organized on an hourly basis for each considered carrier. In this format, every field of a message depicts an amount of energy by which the forecasted profile should be modified in the appropriate time frame to achieve a certain effect. Focusing on values for electric energy, the defined message given in the aforementioned table requests to increase the load during the mid-day period between 11:00 and 14:00 (this is the time of day when peak production from PV panels can be expected), while also encoding the reduction of demand values during the afternoon period between 17:00 and 21:00.

**Table 1.** An example of an explicit DR message for load adjustment.

| Hour of Day | 0 | 1 | 2 | 3 | 4 | 5 | 6 | 7 | 8 | 9 | 10 | 11 | 12 | 13 | 14 | 15 | 16 | 17 | 18 | 19 | 20 | 21 | 22 | 23 |
|---|---|---|---|---|---|---|---|---|---|---|---|---|---|---|---|---|---|---|---|---|---|---|---|---|
| electric [kWh] | 0 | 0 | 0 | 0 | 0 | 0 | 0 | 0 | 0 | 0 | 0 | +2.25 | +1.5 | +1.25 | 0 | 0 | 0 | −1.3 | −0.8 | −1 | −1.8 | 0 | 0 | 0 |
| thermal [kWh] | 0 | 0 | 0 | 0 | 0 | 0 | −5 | −8 | −3 | −4 | 0 | 0 | +6 | +8.5 | +9.5 | +3 | 0 | 0 | 0 | 0 | 0 | 0 | 0 | 0 |

Depending on locally available RES installations, typical user behaviour, forecasted profiles, and other relevant constraints, this format can be utilized to achieve various effects in terms of load reshaping. However, the optimization model that will be described in more detail in the following sections natively takes into account all factors that affect cost-effectiveness such as utilization of available RES generation as well as variable pricing. Therefore, in the context of the proposed platform, explicit DR messages can be more effectively utilized as means of enforcing demand changes that go against the factors that naturally exist. For example, this may be the case when there is a planned outage or necessary infrastructure maintenance and in these cases a load reduction event can be scheduled beforehand at the appropriate time to help reduce the demand levels. For the example given in Table 1, specifically the electric domain, the corresponding requested load modifications are visualized in Figure 6a, and the resulting desired aggregate electric load profile with these modifications incorporated are shown in Figure 6b.

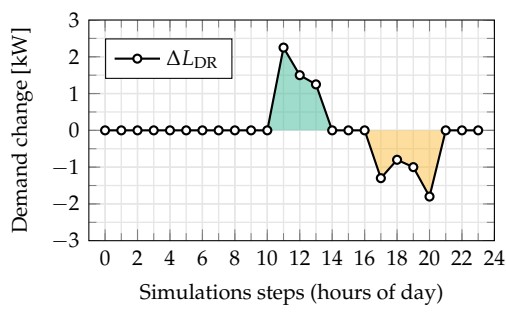

(**a**) DR message (elec.)

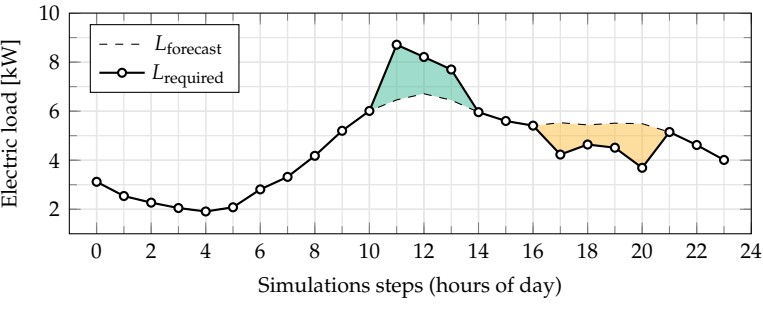

(**b**) Resulting ideal aggregate load modifications

**Figure 6.** An example of an explicit DR event defined for electric power of multiple residential consumers (orange signifies energy to be removed wile green signifies energy to be added) and the ideal resulting aggregate profile.

Formally speaking, the defined DR messages are converted into load modifications represented by a specific variable $\Delta L_{\text{DR}}$. In combination with the forecasted demand profile $L_{\text{forecast}}$, it defines the required load profile $L_{\text{required}}$ as given by

$$L_{\text{required}}(k) = L_{\text{forecast}}(k) + \Delta L_{\text{DR}}(k).$$

This required profile will be later exploited in the Optimization block where the optimal load profile will be penalized for deviating from the required one, as described in the following section.

### 4.4. Optimization

The Optimization block is aimed at converting forecasted energy production and demand and custom grid-related requests into an optimal demand curve for an entire neighborhood. This curve can later be used to generate both non-user-specific and user-specific suggestions to modify the demand aiming at mitigating potential problems with grid stability and responding to requests from a (virtual) DR aggregator. Namely, it takes into consideration day-ahead energy prices (collected in the Measurement block), the forecasted renewable production and the predicted demands from individual users (both of them generated in the Forecasting block) aggregated into a neighborhood profile as well as load modification requirements (given in the DR message generation block). Using supposed demand flexibility, the optimizer shifts the demand in intensity and in time to generate a profile that is the most cost-effective for dwellers and most stable for the grid operator.

This optimization model is developed upon the core constraints that govern the way the Energy Hub is used to model energy transmission and transformation, as described in [54]. The corresponding MILP model that is used for the optimization is defined by laying out all the constraints and bounds in an appropriate matrix form such as given by

$$A_{eq}x = b_{eq} \quad \wedge \quad A_{ineq}x \leq b_{ineq} \quad \wedge \quad l_b \leq x \wedge u_b,$$

with an objective function defined as $J = f^T x$. Since the proposed architecture of the RESPOND system regards the demand as an aggregated value equal to a sum of individual consumption of different dwellers, the demand is also managed and optimized in an aggregated form. In order to do so, and considering that in energy management solutions that employ DR, either the required demand curve is known before hand or that the modifications that should be made to the demand profile are given some time ahead, a key variable defined as the demand deviation $\Delta L$ is given by

$$\Delta L(k) = L_{required}(k) - L(k),$$

where $L$ is the optimized (output) demand and $L_{required}$ is the profile that is desirable to achieve. The demand variable $L$ is limited to values in the range of the forecasted value $L_{forecasted}(k)$ plus/minus a predefined flexibility margin which is (for the following use case demonstrations) set to be 20% of the forecasted value at the considered time stamp. This elasticity band is used to model potential demand flexibility that can be achieved with user interaction and suggestions as given by [55], but it can also be adapted to different user habits if they display more complex behavioral patterns. With the main goal of the optimization being to provide a demand curve that is as close as possible to the required one while ensuring cost-effectiveness for end users, the objective function is constructed as a linear combination depicting monetary parameters (import energy costs and export energy rebates) and penalized values of these demand deltas. In order to do so, its values must be split into positive instances $\Delta L^+(k)$ and negative instances $\Delta L^-(k)$, that is,

$$\Delta L^+(k) \geq 0 \quad \wedge \quad \Delta L^-(k) \leq 0,$$

so that each one of these variables can be penalized in the criterion function with positive and negative values, respectively. This is achieved by redefining the demand delta as

$$\Delta L(k) = \Delta L^+(k) + \Delta L^-(k) = L_{required}(k) - L(k).$$

However, in order to force these newly introduced values to equal positive and negative deviations, a set of constraints is introduced with

$$\Delta L^+(k) \leq +I(\Delta L^+(k)) \cdot \Delta L^+_{\max}(k)$$
$$\Delta L^-(k) \geq -I(\Delta L^-(k)) \cdot \Delta L^-_{\max}(k),$$

that limits these variables to the absolute maximum positive deviation $\Delta L^+_{\max}(k)$ and maximum absolute negative deviation $\Delta L^-_{\max}(k)$, but also introduces the indicator variables $I(\Delta L^+(k))$ and $I(\Delta L^-(k))$ that should equal 1 only if the appropriate deviation exists in time step $k$. Furthermore, the previously mentioned maximum deviations are determined based on the maximum demand flexibility while the indicator variables are forced to take values from the limited set of {0, 1} by denoting them as Boolean variables in the MILP model and also adding a constraint

$$I(\Delta L^+(k)) + I(\Delta L^-(k)) \leq 1.$$

Finally, in order not to allow the engine to reduce all demands to the minimum possible value, an integral constraint is added with

$$\sum_{k=k_1}^{k_2} L(k) = \sum_{k=k_1}^{k_2} L_{\text{forecast}}(k),$$

where $k_1$ and $k_2$ represent the beginning and ending indices of a sliding window for the optimization. With the limits of these windows set to values which are 24 h apart, this constraint essentially ensures that the total energy spent on a daily basis in accordance with the optimal profile is the same as would be the case with the forecasted profile. In other words, the strength of load modifications is implicitly limited to intra-day reallocation.

The criterion of the optimization problem is constructed as a weighted sum with two parts, one corresponding to the operational costs for end users (i.e., difference between the import revenue and export profits) and penalty for the load deviations (in accordance with previously defined DR events). Concretely, the criterion is formulated as

$$J = \sum_i \sum_k (\alpha_i(k) P_{\text{in}}(i,k) + \beta_i(k) P_{\text{exp}}(i,k)) + \sum_i \sum_k (w^+_{\text{DR}} \Delta L^+(i,k) + w^-_{\text{DR}} \Delta L^-(i,k)),$$

where $\alpha_i(k)$ and $\beta_i(k)$ denote the instantaneous energy prices for imports ($P_{\text{in}}(i,k)$) and exports ($P_{\text{exp}}(i,k)$) of energy carrier $i$ at time step $k$ while $w^+_{\text{DR}}$ and $w^-_{\text{DR}}$ are the previously mentioned positive deviation and negative deviation penalization factors used for including the DR event in the criterion. By setting different values of $w_{\text{DR}}$ parameters, the importance of the load's adherence to the profile requested by explicitly defined DR events can be weighted against the minimization of costs for the neighbourhood, and reevaluated based on each individual application use case. As can be observed by its structure, this criterion is formulated as a linear combination between a set of constants and variables, and as such can be optimized using well-known and efficient state-of-the-art algorithms for these types of MILP problems with their essentials described in [56] and implemented as, for example, IBM's ILOG CPLEX (https://www.ibm.com/products/ilog-cplex-optimization-studio, accessed on 15 March 2021) as the arguably most prominent commercial solution or COIN-OR's open source branch-and-cut (CBC) solver (https://www.coin-or.org/, accessed on 15 March 2021).

With a set of miscellaneous constraints, the aforementioned set of equations and the given objective function form a MILP optimization problem implemented by the RESPOND approach. An example of demand reallocation that can be achieved with the help of the optimization will be given in Section 5.

*4.5. Control*

Once the optimal energy profile is generated for each neighborhood, some specific control actions need to be performed by the dwellers in order to achieve such a profile. It has been demonstrated that the potential in the flexibility of appliances' operation and time of use allows them to be exploited for matching the needs of specific DR programs [57]. Therefore, the main goal of the Control block is to translate the optimal energy demand profile into specific DR events mainly related to the scheduling of appliances.

Balancing energy efficiency and user satisfaction is another unresolved DR challenge [58], therefore, the proposed corrective actions are aimed to generate the least disturbance in the every-day operations of dwellers. However, generally, consumers are not willing to change their habits and daily routines in order to enable smart appliance operation and they want to be able to retain full control over their appliances if desired [59]. To avoid this adoption barrier, dwellers specify a set of preferences with regards to the different events and situations involved in DR events. These preferences include:

- The type of actions allowed by each dweller (e.g., switch on or switch off);
- The appliances upon which these actions may be performed (e.g., a dishwasher or a washing machine);
- The periods of time when these actions may be allowed (e.g., from 19:00 onwards);
- The type of notifications preferred by dwellers (e.g., informative, prescriptive or none of them).

The set of preferences for each dweller are specified in the RESPOND mobile app and stored in the MySQL database. Since dweller preferences may be subject to changes throughout time, they may easily modify them whenever they want in the RESPOND mobile app too.

The Control block is aimed at translating the optimal profiles generated from the Optimization block into the actual control actions. For the electric domain, this block has focused on generating the scheduling of appliance usage (e.g., turning the dishwasher on at a certain time) implementing an heuristic optimization algorithm. More specifically, the tabu search (TS) algorithm is used, which aims to solve combinatorial or non-linear optimization problems through memory and so called tabu restrictions [60]. The actual actions proposed by this heuristics service are guided by the preferences set for each dwelling, specially the periods of time when the applications may be activated and they can be both user-specific or mass recommendations for all the neighbors.

The goal of TS is to get closer to the optimal solution while avoiding to get locked in local optima. To do so, TS uses memory to store old movements, prioritizing other kind of movements (similar solutions) that allows the algorithm to reach different solutions from the solutions search space. The algorithm iterates through several solutions ($X$), moving from one solution $s$ to another solution $s'$ from the neighborhood system $N(s)$ of $s$. The new movements aim to improve the current best solution evaluating their objective function value ($f(s)$), in order to minimize the mentioned best value.

The actual scheduling problem can be mathematically represented as a combinatorial optimization problem, where there is a discrete solution search space, that is, the number of possible solutions or feasible schedules is finite. The objective is to minimize the sum of the difference between the optimal demand profile ($o_k$) and the total demand ($r_k$) at each moment of time ($k$), that is, to adapt the use of appliances in such a way that the aimed demand (consumption of all appliances ($p_{i_u}$) plus the fixed demand ($z_{k,u}$) of each dwelling ($u$)) is as close as possible to the optimal demand:

$$\min_{X} f_{\text{objective}} = \sum_{k=1}^{K} \left| \left( \sum_{u=1}^{U} \left( \sum_{i_u=1}^{I_u} \left( p_{i_u} \times x_{i_u}^{k} \right) + z_{k,u} \right) - o_k \right) \right|.$$

The solution is represented as a $X_{I_u,K}$ matrix where $I_u$ is the number of appliances from $u$ dwelling and $K$ is the number of moments of time represented. Each variable $x_{i_u}^{k}$ from the solution matrix is set to 1 if $i_u$ appliance should be on, and 0 otherwise.

The algorithm tries to fulfill several restrictions related to consumption limits for each dwelling, and also about the solution representation (i.e., how to build a solution by 1's and 0's). An example of the underlining appliance schedule is illustrated in Figure 7a, which depicts how the algorithm attempts to achieve a profile close to the optimal one by combining fixed demand with shiftable appliances. Once the recommended control actions are calculated, they are sent to the dwellers in the form of a mobile app notification, as it can be seen in Figure 7b.

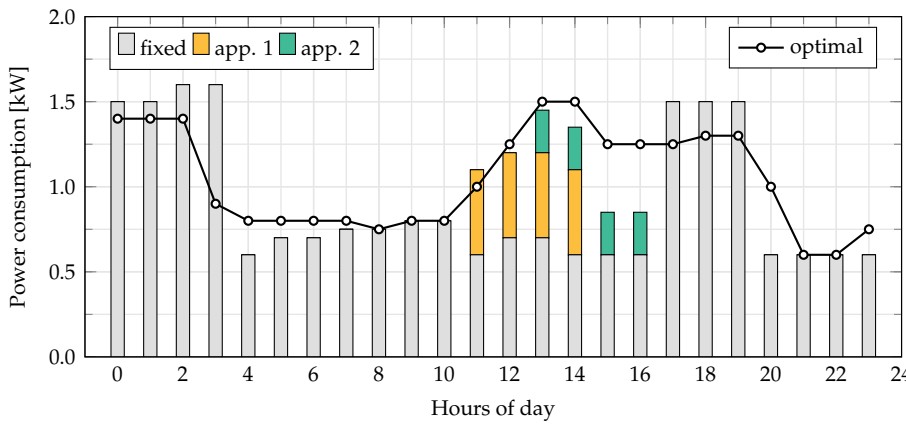

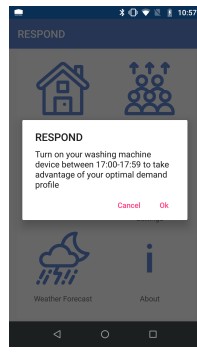

(**b**) In-app notifications

(**a**) Optimization of appliance activations with regards to the load curve

**Figure 7.** An example of the optimization process for appliance activations and resulting notification

## 5. Use Case Demonstration: Implicit DR in Madrid

The main use case that will be analyzed to determine to what extent the dwellers can respond to different conditions and display flexibility in their power consumption, is the application and effects of price-based implicit DR events that were implemented in the Madrid pilot site. Namely, the final two years of the RESPOND project have been split into two periods, the baseline period (1 November 2018–31 October 2019) and the validation period (1 November 2019–31 October 2020). The former was used to determine the reference behaviour of the dwellers in terms of energy consumption (typical demand profile, typical total consumed energy, etc.). During the latter period, the dwellers that have signed contracts with Fenie Energia have been switched between different experimental pricing tariffs and changes in their behaviour were recorded.

With different dwellings being contracted to different plans and each one having its own higher electricity price (P1) and lower electricity price (P2) values, the corresponding prices for the analyzed dwellings, applicable for both the baseline and validation periods, are given in Table 2. During the baseline period, the dwellings considered in this analysis were contracted to ToU tariffs that follow the shape illustrated in Figure 5a with appropriate P1 and P2 values from Table 2. On the other hand, Table 3 shows how the pricing tariffs have varied only during the validation period. Furthermore, different cases employ different periods of the day when energy is offered for free. Although minor differences between price values for pilot dwellings do exist, since all the dwellings are subject to the same tariff profile from Table 3 at a given time, the differences can be neglected when considering all dwellings jointly.

**Table 2.** Electric energy prices for experimental tariffs in [EUR/kWh].

| Household Codename | P1 (w/o tax) | P2 (w/o tax) | P1 (w/ tax) | P2 (w/ tax) |
|---|---|---|---|---|
| M1 | 0.157 | 0.086 | 0.167 | 0.091 |
| M3 | 0.157 | 0.082 | 0.167 | 0.087 |
| M0, M2, M4, M6, M10, M12 | 0.149 | 0.078 | 0.158 | 0.083 |

**Table 3.** Different experimental pricing tariff profiles.

| Tariff | Start | Finish | 00-12 | 12-13 | 13-15 | 15-16 | 16-17 | 17-22 | 22-23 | 23-00 |
|--------|-------|--------|-------|-------|-------|-------|-------|-------|-------|-------|
| Case 1 | 2019-11-01 | 2020-03-31 | 0 | P1 | P1 | P1 | P1 | P1 | 0 | 0 |
| Case 2 | 2020-04-01 | 2020-04-26 | 0 | 0 | P1 | P1 | P1 | P1 | P1 | 0 |
| Case 3 | 2020-04-27 | 2020-05-31 | P2 | P2 | P1 | 0 | P1 | P1 | 0 | P1 |
| Case 4 | 2020-06-01 | 2020-08-31 | P2 | P2 | P1 | 0 | 0 | P1 | 0 | 0 |

The first two tariff cases (Case 1 and Case 2) have not resulted in major modifications to the load profile as the periods with free energy correspond to periods when energy was also cheaper before. However, this had prompted the inclusion of the latter two tariff cases (Case 3 and Case 4) where the dwellers have been offered the so-called "happy hours" or, in other words, periods during the afternoon when they can consume energy for free. This experiment was designed specifically with the goal of providing a notable incentive for displayed demand flexibility, and therefore, testing if such flexibility can be induced. With the Case 3 tariff initially offering two short one-hour-long periods with free energy, it was later extended to include two two-hour-long "happy hour" periods, as defined by Case 4 tariff. This case will be specifically analyzed in the following paragraphs to illustrate the observed effects of implicit DR.

In order to provide the data of the best quality for both baseline and validation periods, logs obtained by the RESPOND system have been employed to extract hourly electric energy consumption. The resulting values have been manually curated so that only day-long logs with continues values remain, meaning that the days with periods in which the corresponding measurements do not follow the typical demand curve (e.g., values are constant or only display white noise) have been removed. By doing this, only the days when the dwellers have been actively using their appliances have been considered. Given the length of the different tariff cases during the validation period, it was decided that the best analysis could be provided by comparing the same month of two different years. A month-long period is considered long enough to provide enough data for meaningful analysis, but still short enough so that the expected behaviour of the dwellers can be comparable, given the presumed similar average meteorological conditions, holidays, and so forth. Therefore, the baseline month for in depth analysis was selected to be June 2019 while the validation month was selected to be June 2020. The resulting data contains the following number of day-long sets of hourly energy consumption: 132 for the baseline period and 241 for the validation period, further separated into working days and nonworking days (weekends or public holidays). Since related literature generally regards working days and nonworking days as different categories for energy consumption analysis, and since working days were more frequent in the curated data, the use case that will be elaborated next will focus only on data corresponding to working days.

The extracted values are evaluated jointly for all dwellings, meaning that if the extracted data were to be considered as a table, each row would correspond to one day-long record for a given house on a given day, while the columns would correspond to hours of a day. In order to illustrate the distribution of total hourly energy consumption, a number of metrics have been evaluated. Let $\mathcal{L}_k$ denote the sample distribution of individual data observations $x$ that corresponds to consumption between $k$ and $k+1$ hours and $F_{\mathcal{L}_k}(x)$ denote the corresponding cumulative distribution function (CDF). If defined as such, observations of $\mathcal{L}_k$ are given in column $k$ of the data table. Then, for each $k$ a set of quartiles is calculated using the following definition

$$q_n(\mathcal{L}_k) = p \quad \Longleftrightarrow \quad F_{\mathcal{L}_k}(x) = P(x \leq p) = n \cdot 25\%,$$

where the ordinal constant is chosen as $n \in \{1, 2, 3\}$ and therefore corresponds to the 25th, 50th (median) and 75th percentile. Figure 8 illustrates how these metrics change on an hourly basis for the chosen baseline month and validation month. The first observation that can be made is that the difference in tariffs appear to have made a significant impact

on the shape of the third quartile of the consumption curves. In other words, the two peaks between 12:00 and 15:00 and between 19:00 and 22:00 observable for the $q_3(t)$ curve in Figure 8a appear to have been substituted by a single, more pronounced peak, in the corresponding curve from Figure 8b between 15:00 and 17:00, exactly when the "happy hours" are placed. In addition, by subtracting the two curves that depict median hourly consumption

$$\Delta q_2(k) = q_{2\text{-valid}}(k) - q_{2\text{-base}}(k)$$

set of median consumption deltas is obtained and depicted in Figure 9a. As is clearly shown here, the dwellers exhibited a clear bias towards lower energy consumption during the validation period when compared to the baseline one with the average median consumption delta equaling $\overline{\Delta q_2} = -82.2$ W. However, by centering the median consumption deltas with the average median consumption delta

$$\Delta q_{2\text{center}}(k) = \Delta q_2(k) - \overline{\Delta q_2} = \Delta q_2(k) - \frac{1}{24}\sum_{k=1}^{24} \Delta q_2(k),$$

a set of centered deltas is obtained and is illustrated in Figure 9b. Here it can be observed that the most notable delta when compared to the average are between 14:00 and 17:00 which corresponds to the "happy hour" period. As was expected, this increase in consumption is not a result of the need for additional energy (which is also clear from Figure 8a) but rather the temporal relocation of existing consumption, with the periods mostly affected by overall demand decrease between 12:00 and 15:00.

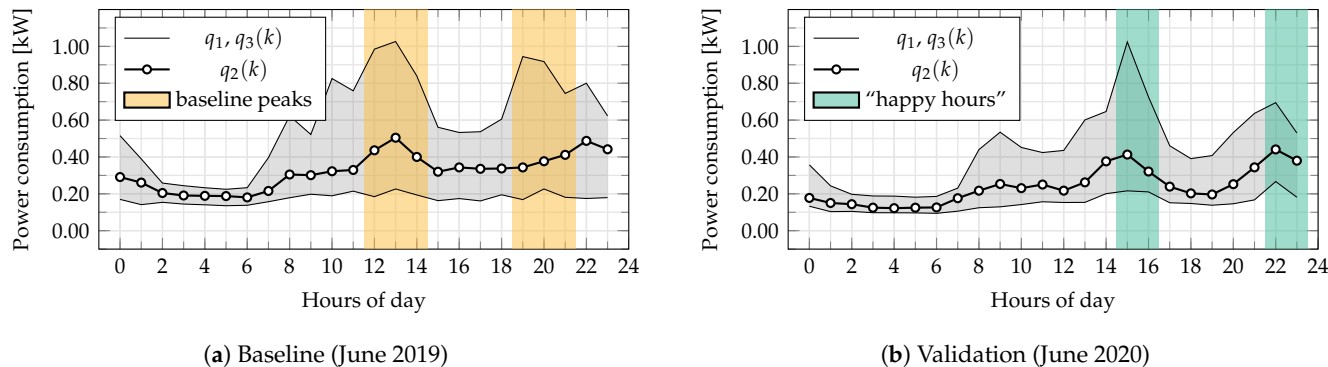

(**a**) Baseline (June 2019)          (**b**) Validation (June 2020)

**Figure 8.** Comparison between the distributions of hourly energy consumptions.

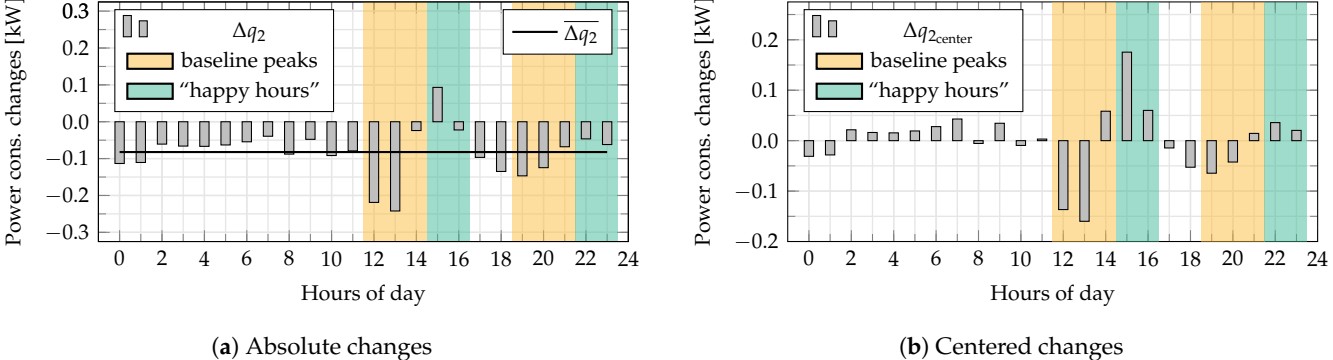

(**a**) Absolute changes          (**b**) Centered changes

**Figure 9.** Deltas between median hourly electric energy consumption.

Going back to the absolute changes from Figure 9a, even though, on average, the hourly consumption during the validation month is lower, the difference is positive at $\Delta q_2(15) = 93.2$ W between 15:00 and 16:00, further illustrating the notable effects that are made to the demand allocation by the introduction of the "happy hours" with free energy.

However, observing and analyzing absolute changes in power consumption between different periods of time usually requires much more complex means of normalization that would take into account different factors that influence the overall consumption such as meteorological conditions. As the available data in this regard is relatively limited, a much more informative analysis can be conducted when observing the differences in demand profiles. In order to do this, the ratio of median energy consumed for each hour with respect to the total daily consumption is calculated as

$$q2_{\text{prof-}i}(k) = \frac{q_{2-i}(k)}{\sum_{k=1}^{24} q_{2-i}(k)},$$

for $i \in \{\text{base}, \text{valid}\}$ and is depicted in Figure 10. As this graph clearly shows, during the nighttime and early morning periods, where there are no modifications to the tariff, no significant differences can be observed between the demand profiles.

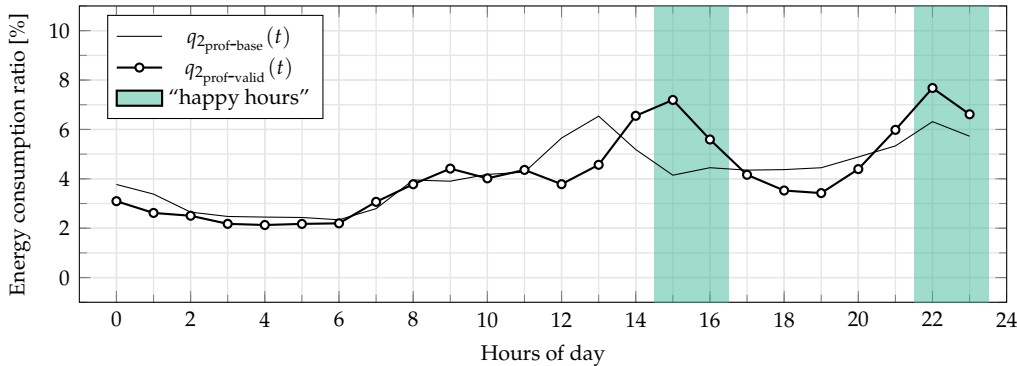

**Figure 10.** Visual comparison between hourly demand profiles.

However, a significant shift is to be noted for the peak period that was between 12:00 and 14:00 and appears to have moved to between 14:00 and 16:00 during the validation month. Furthermore, a slight decrease of consumption between 16:00 and 20:00 is also evident during the period with higher prices, as well as an increase in the allocated demand ratio between 22:00 and 00:00 which, again, corresponds to hours with free energy. Table 4 shows a section of most notable demand profile values as well as their absolute and relative differences defined as

$$\Delta q2_{\text{prof}}(k) = q2_{\text{prof-valid}}(k) - q2_{\text{prof-base}}(k) \quad \text{and} \quad \delta q2_{\text{prof}}(k) = \Delta q2_{\text{prof}}(k) / q2_{\text{prof-base}}(k).$$

These values once again illustrate the impact that is made by the "happy hours" on the shape of the energy profile, but also show that, given proper incentives, dwellers can display significantly more flexibility than was originally presumed.

**Table 4.** Most notable differences between hourly demand profiles.

| Variable | 12-13 | 13-14 | 14-15 | 15-16 | 16-17 | 18-19 | 19-20 | 22-23 | 23-00 |
|---|---|---|---|---|---|---|---|---|---|
| $q2_{\text{prof-base}}(k)$ [%] | 5.65 | 6.54 | 5.19 | 4.15 | 4.45 | 4.37 | 4.45 | 6.32 | 5.72 |
| $q2_{\text{prof-valid}}(k)$ [%] | 3.78 | 4.57 | 6.55 | 7.19 | 5.59 | 3.53 | 3.42 | 7.68 | 6.61 |
| $\Delta q2_{\text{prof}}(k)$ [%] | −1.87 | −1.97 | +1.37 | +3.05 | +1.14 | −0.85 | −1.03 | +1.36 | +0.89 |
| $\delta q2_{\text{prof}}(k)$ [%] | −33 | −30 | +26 | +74 | +26 | −19 | −23 | +22 | +16 |

## 6. Conclusions

With buildings accounting for more than a third of global energy use, the implementation of proper operation strategies has shown great potential in terms of achieving significant energy savings. The residential sector is specially promising, as it is responsi-

ble for around the 25% of energy usage. Furthermore, DR can influence customers' use of electricity in effective ways, but their implementation in the residential sector is not straightforward. In this regard, RESPOND aims to bring DR programs to neighbourhoods across Europe by solving the solving the energy demand optimization considering the management of collectively shared RES generation as well as variable pricing tariffs, specific demand flexibility constraints and dwellers comfort.

Achieving such a multi-objective problem is complex and to that end, RESPOND proposes an AI system that comprises five blocks—Measurement, Forecasting, Demand Response, Optimization and Control. The Measurement block aims at acquiring and storing all the necessary energy, ambient and other relevant data. The Forecasting block focuses on exploiting this data to forecast the short-term generation of PV panels and STCs as well as the energy demand at a dwelling and neighborhood level. The Demand Response block is responsible for generating the messages that request load modifications. This information is later on used by the Optimization block to calculate the optimal demand curve. Finally, the Control block generates and suggests the adequate corrective actions to dwellers in order to achieve this optimal demand curve while minimizing the disruption of their daily habits.

The implementation of this AI system has been demonstrated in a real-world use case involving electric loads and implicit DR. The benefits of using such a system are two-fold. The dwellers are informed of what is the best way to make use of lower energy prices and local generation while the grid has been offered a notable level of demand flexibility. In the discussed study, two key findings have been observed: on the one hand, with a ToU tariff that is designed in accordance with the schedule that the residents are already used to, even with extremely low prices, no major changes to the load profile should be expected. On the other hand, when a notable incentive is offered in contrast to the existing habits, such as free energy in two-hour windows adjacent to afternoon peak periods, the dwellers have shown the willingness to optimize and shift their load in line with newly defined "happy hours". Furthermore, it also demonstrated that, given the proper incentives, dwellers can display significantly more flexibility in terms of load elasticity than was originally often presumed. Analyzing the daily energy share used on an hour-by-hour basis has indicated that previous peak periods exhibit a decrease of around one third (e.g., energy spent between 12:00 and 13:00 dropped from 5.6% to 3.8% of the daily total) while "happy hours" saw an increase in energy share of almost double (e.g., energy spent between 15:00 and 16:00 increased from 4.2% to a notable 7.2% of the daily total). These results show a high potential of temporal load reallocation for further developments of DR programs in the residential sector.

**Author Contributions:** Conceptualization, I.E.-G., M.J., D.P., F.J.D. and N.T.; formal analysis, I.E.-G., M.J. and D.P.; funding acquisition, F.J.D. and N.T.; investigation, I.E.-G., M.J. and D.P.; methodology, I.E.-G., M.J. and D.P.; project administration, F.J.D. and N.T.; validation, I.E.-G., M.J. and D.P.; writing—original draft preparation, I.E.-G., M.J. and D.P.; writing—review and editing, I.E.-G., M.J., D.P., F.J.D. and N.T. All authors have read and agreed to the published version of the manuscript.

**Funding:** The research presented in this paper is partly financed by the European Union (H2020 RESPOND project, Grant Agreement No.: 768619), the SPRI-Basque Government through the ELKA-RTEK program (3KIA project (KK-2020/00049)), as well as the Ministry of Education, Science and Technological Development and the Science Fund of the Republic of Serbia (AI-ARTEMIS project, #6527051).

**Informed Consent Statement:** Informed consent was obtained from all subjects involved in the study.

**Acknowledgments:** The authors would like to acknowledge the support that was provided by their colleagues from the team at Feníe Energía lead by Antonio Colino, Rodrigo Lopez and Agustina Yara.

**Conflicts of Interest:** The authors declare no conflict of interest. The funders had no role in the design of the study; in the collection, analyses, or interpretation of data; in the writing of the manuscript, or in the decision to publish the results.

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
