# Peer review of "An AI-Powered System for Residential Demand Response"

_electronics, doi:10.3390/electronics10060693_

Round 1

Reviewer 1 Report

I find the paper of interest to the journal. The structure and sections of the paper are well distributed. The title and abstract are in line with what is reported in the conclusions. I am in favour of its publishing. 

Author Response

Answer to reviews provided in the attached document

Reviewer 2 Report

Thia paper uses artificial intelligence to solve the complex multi-objective problem to bring demand response programs to the residential sector.

It states the main contributions of the article as follows:
• The development of an AI-powered system for demand response in residential buildings; (Where are the AI details? There is missing information  on the learning mechanism nor the codes for a NN)
• The application of the AI-powered system in a real-world neighbourhood; (Missing details on the neighnourhood and how the AI was applied)
• Decrease of energy demand peaks by above 30%; (this is not the research contribution - it could be that the system may provide some decrease in energy demand - how can the authors assure minimum of 30% is not clear.  maximum is how much?  This range could be an objective or target to achieve  but not a contribution)
• Increase of energy demand in non-peak periods by nearly 50%;(same as above)

• Assessing the changes to the load profile with regards to discounted energy prices.(This is an activity and not the research contribution)

The authors are required to address the comments above in italics and include in the revised paper, a thorough in-depth information.

Author Response

(The authors gave the same response as above.)

Reviewer 3 Report

  The paper is interesting, but the mathematical description of the used algorithms is missing. The mathematical description of the applied procedures should be introduced in the manuscript. For example, the algorithm applied for the "fobjective" function minimization has to be mathematically described. Also, the used AI algorithms have to be mathematically presented.

Author Response

Answer to reviews provided in the attached document

This manuscript is a resubmission of an earlier submission. The following is a list of the peer review reports and author responses from that submission.

Round 1

Reviewer 1 Report

This paper presents the technical content of the RESPOND project, that aims to study coordinate, trigger and analyse electric load flexibilities from end-users. The paper first presents the overall approach (measurement, forecast, communication of DR effort required, optimization, communication with dwellings for actual control), then describes the content of each phase of the process, and finally describes the results of a real implementation during 2 years with real dwellings.

The strength of the paper mostly lies in the platform description that is exhaustive, with a mention of different technologies used, and in the fact that it provides a feedback of real houses flexibility and ability to shift their demand, which is scarce.

However, the reviewer belives this paper can be improve in many ways. Comments are listed below, following the structure of the paper:

  1. Title: it is good to have the word "AI" in the title as it will capture the interest of more readers. However, the description of AI tools used in the study should be emphasized better, and described in more details.
  2. The introduction should really be proof-read, as there are many english mistakes. Amoung other things, authors must check carefully when there is a need to use "the", and when it should be omitted.
  3. It is not clear what problem the Demand Response strategie aims to solve. This is a major point that should be highlighted better. There is a mention of energy portfolio, then grid services. Is there a clear aim, and clear problems that DR is supposed to solve? For example, given the tariffs proposed in Madrid, is there any particular issue they were supposed to solve?
  4. Some acronyms are not defined (SoA, CST-LF, RDF, ...)
  5. Genetic algortihms, PSO, ... are not really heuristics approaches. Population based approaches or nature inspired methods would be a better definition.
  6. When authors use the word "optimal", they should ensure the reader understands what are the evaluation criteria. Does optimal mean "lowest cost", "lowest CO2 emissions", "lowest voltage excursion", a mix of those?
  7. About section 4: it is extremely disturbing to see a "DR message generator" before the optimization. to my understanding, DR is supposed to be triggered to solve an issue or to defend a position on an energy market. This position and the contribution of households come from an optimization. Therefore, I would expect the DR message to be sent after the optimization phase (as it is described in the "translator service"). Therefore, I do not understand what is done in the DR message generator. Fig. 2 should specify "to who" is the DR message sent, as it is unclear. Is there a message sent to end-users to ask them to reduce their load of a certain amount at a particular time? if yes, how this request is computed if it is not computed from an optimization? This is a main issue of the paper and must imperatively be solved.  Also, this reviever believes it is required to specify what concerns the aggregated fleet, and what concerns only individual dwellings. Finally, the settlement phase is not included in Fig. 2, although it should be, at least in order to have a comprehensive vision of the system.
  8. In the measurement, it would be good to specify what was measured: only the dwelling consumption (smart meter) or individual appliances? what type of sensor was used? In Fig,.2, it would be good to display th word "measurement"
  9. ABout the CDM, given the fact that many devices use their proprietary protocol and data model, could you please explain how you ensured that all devices could comply with the CDM? did you develop a translator between each service/device and the MQTT topic? Also, what were the communication protocols used (TCP/IP, Zigbee and then internet, ...?)
  10. Until line 279, it is not clear what is used where (ML in Aarhus & Madrid), Physics based model in Aran
  11. MPAE should be replaced by MAPE.
  12. It should be specified what is the time wondow that was used for the MAPE computation. Also, the maximum error in absolute value could be of interest. Finally, the authors should explicitly mention the level of aggregation (in terms of number of housholds and time (1h forecast I suppose)) for the forecasts that were used for MAPE computation
  13. The sentence in line 304 looks contradictory with the reviewer's experience, but I am only mentionning this, not requesting extra work: indeed, when using ML methods, it seems that forecasting a city's consumption using the aggregated data gives more accurate results than adding the forecast of every dwelling's forecast. 
  14. The paper's title mentions that it focuses on AI methods. AI methods are mentioned, but not presented in detail, which could be a great added value to the work. Therefore, the reviewer would recommend to explain how the inputs are used (how is the hour included as an input, how are the type of day included (0 -1, or something else?), how is the output of the solution ? is it dirctly the consumption, normalized, ...? 
  15. For the demand response messages subsection, it is unclear what are the recipients of these messages. Can you confirm in the text that we are considering residential users? Also, it would be interesting to precise around line 337 what scheme is used where.
  16. In Fig. 6, what does it relate to ? one dwelling?? an industrial load?
  17. How is Lrequired computed (line 392)? does it come from an optimization? 
  18. Line 403: what is a "supposed flexibility"? how is flexibility assessed?
  19. line 405: it is necesary to clarify the link between the profile that is generated here, and the DR message. It is not clear what is embedded in the DR message. I would have expected that this profile was embedded within the DR message.
  20. The authors should provide a clear formulation of the MILP optimization problem: min f(x), such that ....
  21. How is the ToU included in the optimization formulation?
  22. Why is Lreal = Loptimized and not Lreq ? this looks weird (line 414).
  23. Th fixd flexibility margin of 20% should be discussed maybe, as it depends on the type of appliances. someone with electric heating could have much more flexibility margin
  24. Equations line 426: authors should add the non negativity/non positivity inequalities
  25. Line 434 (Equation): is Lforecast (line 392)the same as Lpredicted 
  26. Line 488: the authors should precise how the preferences are used in the constraints of the optimization problem
  27. For the use case: Table 3: it is not clear why the authors include Case 1, 2 and 3, especially as it is not used in the analysis. The reviewer perfectly understands what these cases are, but maybe it would be better to avoid mentioning them as they add too much information (not required) to the readers
  28. Line 550: Is it a distribution for one dwelling based on the consumption over the whole month? It is difficult to see the data that is considered (of an individual dwelling of of the whole fleet of households, over one month, one day? the definition of the distribution should be better explained)
  29. Fig 9: what does the yellow highlight represent? the peak? the green highlight represents either the peak or the price change (which corresponds to the same in this case), but for yellow color, it would be good to add a legend. Also, Fig. 9: is the data normalized? does it show the data from one house only (averagre of every day's data for the whole month), or is it an average over all houses over the whole month?
  30. The authors should also explain to the reader what are the gains (at least financial gains) for the end-users, and what it brought to the aggregator or grid (i.e. what problem was DR supposed to solve, and did it solve it?)

Reviewer 2 Report

Different scenarios should be considered to demonstrate the effectiveness of the proposed solution.

Authors could highlight the contribution and novelty of the presented work more effectively in introduction section. 

Reviewer 3 Report

This paper deals with assessing the feasibility of KNN and MILP-based demand response for residential energy management. To improve the paper quality, the following issues need to be addressed as follows.

  • In this paper, the combination of K-NN and MILP was proposed to address residential energy management, where K-NN was applied to predict the energy demand of residents, and MILP was used to optimize the energy cost of residents. However, it is not appropriate to declare that the adopted method is an artificial intelligence-based method. The declaration should be more precise in reflecting the paper's contribution.
  • The contributions of this paper were presented in a whole paragraph, which could not highlight the novelty of this paper appropriately. It is suggested that the contributions of this paper should be further summarized point by point.
  • The author simply claims that several machine learning methods are used to predict the residents’ energy demand. Then, it suddenly declares that a k-nearest neighbors (k-NN) algorithm can obtain the best performance. This description is not convincing. To make the paper more readable when including many prediction approaches, please consider to include a table to summarize the comparison results (i.e., the name, the inputs, the architecture, the training platform, the computation time, and the prediction accuracy of each algorithm, etc.)
  • What is the advantage of using the proposed MILP method compared to the current popular reinforcement learning approaches?

In summary, the authors should elaborately describe the contribution to increase the novelty, and carefully present the mentioned prediction methods in DR and explain what distinguishes K-NN from the others.

Round 2

Reviewer 3 Report

This paper has been revised, but it does not seem to respond to my major concerns.

This paper has been revised, but it does not seem to respond to my major concerns.

The terms “artificial intelligence (AI)” is often used to describe machines (or computers) that mimic "cognitive" functions that humans associate with the human mind, such as "learning" and "problem-solving" [1]. In other words, an AI-based system is capable of self-learning and self-adjusting to achieve its desired objective in a dynamic environment. In the manuscript, it is hard to capture how the author reflects the characteristics of AI. Hence, it is not appropriate to declare that the adopted method is an artificial intelligence-based method.

[1] Russell, Stuart J.; Norvig, Peter (2009). Artificial Intelligence: A Modern Approach (3rd ed.). Upper Saddle River, New Jersey: Prentice Hall. ISBN 978-0-13-604259-4.

Simply adding a reference (reference 39) still cannot prove that the effectiveness of a k-nearest neighbors (k-NN) algorithm can obtain the best performance in the proposed scenarios